# Cloning of the Maternal Effector Gene *org* and Its Regulation by lncRNA ORG-AS in Chinese Tongue Sole (*Cynoglossus semilaevis*)

**DOI:** 10.3390/ijms23158605

**Published:** 2022-08-03

**Authors:** Xiaona Zhao, Bo Feng, Qian Wang, Lili Tang, Qian Liu, Wenxiu Ma, Chenghua Li, Changwei Shao

**Affiliations:** 1School of Marine Sciences, Ningbo University, Ningbo 315211, China; zxn1328753@163.com (X.Z.); lichenghua@nbu.edu.cn (C.L.); 2Key Lab of Sustainable Development of Marine Fisheries, Ministry of Agriculture and Rural Affairs, Yellow Sea Fisheries Research Institute, Chinese Academy of Fishery Sciences, Qingdao 266071, China; fengbo_1220@foxmail.com (B.F.); wangqian2014@ysfri.ac.cn (Q.W.); tlltgq@126.com (L.T.); liuqian97927@163.com (Q.L.); mawenxiu121@163.com (W.M.); 3Laboratory for Marine Fisheries Science and Food Production Processes, Pilot National Laboratory for Marine Science and Technology (Qingdao), Qingdao 266237, China

**Keywords:** maternal effector genes, oogenesis, *org*, lncRNA, maternal–zygotic transition, Chinese tongue sole

## Abstract

Maternal effector genes (MEGs) encode maternal RNA and protein, accumulating in the cytoplasm of oocytes. During oocyte development, MEGs participate in oocyte meiosis and promote oocyte development. And MEGs can also regulate maternal transcriptome stability and promote maternal–zygotic transition (MTZ) in early embryonic development. Long noncoding RNAs (lncRNAs), as new epigenetic regulators, can regulate gene expression at both the transcriptional and post-transcriptional levels through *cis*- or *trans*-regulation. The oogenesis-related gene *org* is a germ-cell-specific gene in fish, but the role of *org* in embryonic development and oogenesis has rarely been studied, and the knowledge of the lncRNA-mediated regulation of *org* is limited. In this study, we cloned and identified the *org* gene of Chinese tongue sole (*Cynoglossus semilaevis*), and we identified a lncRNA named lncRNA ORG-anti-sequence (ORG-AS), located at the reverse overlapping region of *org*. The results of qRT-PCR and FISH demonstrated that *org* was highly expressed during the early stages of embryonic development and oogenesis and was located in the cytoplasm of oocytes. ORG-AS was expressed at low levels in the ovary and colocalized with *org* in the cytoplasm of oocytes. In vitro experiments showed that overexpression of ORG-AS inhibited *org* expression. These results suggest that *org*, as a MEG in *C. semilaevis*, participates in the MTZ and the oogenesis. The lncRNA ORG-AS negatively regulates the gene expression of *org* through *trans*-regulation. These new findings broaden the function of MEGs in embryonic development and the oogenesis of bony fish and prove that lncRNAs are important molecular factors regulating *org*.

## 1. Introduction

In mammals, the development of early embryos depends on maternal complexes, including subcellular organelles and biomacromolecules stored in the oocyte cytoplasm, encoded by maternal effector genes (MEGs) [1,2]. These maternal complexes participate in oocyte genome transcriptional activation, meiosis, and fertilization, and maintain oocyte growth and development [3]. MEGs were first found in *Drosophila* [4] and then in *Nematode* [5] and *Xenopus* [6]. For example, *Zar1* is specifically expressed in oocytes, and it has been found that the targeted knockout of *Zar1/2* blocks the process of maternal–zygotic transition (MTZ) in humans and mice [7,8]. Researchers also found some germ-cell-specific MEGs in bony fish, such as *vasa*, *nanos1*, and *ziwi* in zebrafish, which can maintain early embryonic development and oogenesis [9,10,11].

Oogenesis is a long process and is the basis of embryonic development, which starts from the differentiation of primary oocytes after mitosis and the proliferation of oocytes [12]. Oogenesis is regulated by a large number of factors in the ovary, including histone modification, nonhistone acetylation, alternative splicing, and other epigenetic modifications, as well as communication between oocytes and their surrounding granulosa cells [13,14]. Ma et al. found that protein deacetylases HDAC1 and HDAC2 in mice can regulate transcription and apoptosis in the development of mice oocytes [15]. In addition, Lu et al. found a large number of differentially expressed genes and differentially alternative splicing genes through a comparative transcriptome analysis of the gonads of Chinese tongue sole (*Cynoglossus semilaevis*), and they proved that alternative splicing, as an important mechanism of epigenetic modification, can regulate reproductive cell development [16,17]. Consequently, epigenetic regulation plays an important role in the development of oocytes.

Noncoding RNAs with a length of more than 200 nucleotides are considered long noncoding RNAs (lncRNAs), which are widely transcribed in eukaryotic genomes with the same 5′ cap structure and 3′ polyA tail as mRNA [18]. According to the position relationship between lncRNAs and adjacent mRNAs in the genome, lncRNAs can be divided into seven categories, namely, sense, antisense, bidirectional, intron, intergenic, promoter, and enhancer lncRNAs [19]. As a new epigenetic regulator, lncRNAs can regulate gene expression by regulating the modification of histones and DNA [20]. In addition, the *cis-* and *trans*-regulation of lncRNA is also important for the regulation of local gene expression [21]. Feng et al. found that lncRNA DMART2-AS is an important regulatory factor of *dmrt2* in *C. semilaevis*, reversing complementation with exon 4 of *dmrt2* and inducing the transcription of *dmrt2* [22]. In the crustacean *Daphnia magna*, lncRNA DAPALR overlaps with *dsx1* 5′UTR and can activate and maintain the expression of *dsx1* [23]. In conclusion, all these findings show the important role of lncRNAs in regulating gene expression.

As an economically important fish in China, the female *C. semilaevis* grows 2–4 times faster than males, and females have a greater economic value [24]. MEGs are crucial for embryonic development and oogenesis in female fish. However, there is no study on the function and regulation of MEGs in *C. semilaevis* at present. Recently, Li et al. found a new gene in zebrafish, *zorg*, a germ-cell-specific gene that is abundant in the oocyte cortex of zebrafish oocytes. *zorg* gene products may play an important role in zebrafish oogenesis and early embryonic development [25]. *org*, as a germ-cell-specific gene only found in fish, has important research value in embryonic development and oogenesis. In this study, we cloned and identified the first MEG *org* in *C. semilaevis*, and we identified the lncRNA, ORG-anti sequence (ORG-AS), that regulates *org* in *C. semilaevis*.

## 2. Results

### 2.1. Cloning and Sequence Analysis of org

We obtained the full-length sequence of the *org* gene using 5′ and 3′ RACE amplification. A sequence analysis revealed that the total *org* cDNA length was 2101 bp, comprising six exons and five introns, and had a poly (A) tail at the 3′ end. The *org* cDNA had a 666 bp open reading frame and encoded a 221 amino acid protein (Figure 1). To further understand the homology and evolution of *org*, we compared the amino acid sequences of *org* with those of other species. We found that *org* only exists in fish and that the amino acid sequence of *org* shares 84.88% homology with other teleost amino acid sequences (Figure 2a). The results of the phylogenetic tree show that *C. semilaevis* formed a clade with *Paralichthys olivaceus*, *Lates calcarifer*, *Seriola lalandi dorsalis*, and *Seriola dumerili*, indicating a relatively close evolutionary relationship (Figure 2b).

### 2.2. ORG-AS Is Predicted to Bind to org

We found that there is a reverse complementary sequence between *org* upstream and a lncRNA ORG-AS in the *C. semilaevis* genome. We hypothesized that ORG-AS might regulate *org*, and we predicted the interaction sites between ORG-AS and *org*. The IntaRNA results showed that 54–203 nucleotides of ORG-AS were inversely complementary to nucleotides 253–402 of *org* (Figure 3a). ORG-AS (nucleotides 86–110) and *org* (nucleotides 348–372) might interact according to the RNAup prediction (Figure 3b). We observed that ORG-AS was inversely complementary to exon 4 of *org* by building the model presented in Figure 3c.

### 2.3. Expression Patterns of org and ORG-AS

We measured *org* and ORG-AS expression levels using qRT-PCR on different tissues in adult *C. semilaevis*, as well as in the early embryos of *C. semilaevis.* The *org* and ORG-AS expression patterns are shown in Figure 4. The results showed that there were high *org* mRNA levels in early embryos and unfertilized eggs. The high expression levels were sustained from unfertilized eggs to the blastocyst stage, and they decreased rapidly in the gastrulation stage. However, the expression of ORG-AS in unfertilized eggs and early embryos was much lower than that of *org*, and the expression of ORG-AS showed a rapid decrease in the gastrulation stage (Figure 4a). In sexually mature *C. semilaevis*, *org* was mainly expressed in the gonads, especially in the ovary, and it was rarely expressed in other tissues. ORG-AS was also mainly expressed in the gonads, but in contrast to *org*, it was highly expressed in testis (Figure 4b).

We further assessed *org* and ORG-AS expression patterns during gonadal development (Figure 4c). The expression of *org* in the ovary increased sharply at 180 dpf, and then it remained stable. Moreover, the expression of *org* was higher than that of ORG-AS in each developmental stage of the ovary. In the testis, the expression of ORG-AS increased rapidly at 180 dpf and was higher than that of *org*, and then it gradually decreased with *org*. Furthermore, in the testis and ovary, ORG-AS was always expressed earlier than *org*.

### 2.4. FISH of org and ORG-AS in Gonads

We investigated the localization of *org* and ORG-AS in oocytes using fluorescence in situ hybridization. In 6 mpf and 1.5 ypf ovaries, there were many somatic cells around the oocytes (Figure 5a,e). The oocytes in the 6 mpf ovaries were mainly phase I oogonia with large nucleoli in the center of the nucleus (Figure 5b). The oocytes in the 1.5 ypf ovaries were mainly phase II and early phase III oocytes, which contain multiple small nucleoli, distributed around the nucleolar membrane (Figure 5f). In 6 mpf and 1.5 ypf ovaries, *org* was mainly localized in the oocyte cytoplasm (Figure 5b,f). In addition, similar to the *org* expression pattern, ORG-AS was mainly localized in both 6 mpf and 1.5 ypf ovaries (Figure 5c,g). Intriguingly, the merged image showed that ORG-AS and *org* were co-located in the cytoplasm of oocyte (Figure 5d,h).

### 2.5. ORG-AS Negatively Regulates the Expression of org

To determine whether ORG-AS negatively regulates *org*, we constructed overexpression vectors for *org* and ORG-AS and then transfected 293T cells (Figure 6a). The qRT-PCR results showed that compared with the *org* transfection groups, *org* expression decreased significantly in the ORG-AS and *org* co-transfection groups. While the expression of *org* in the ORG-AS and pcDNA3.1 co-transfection groups did not significantly change (Figure 6b). The results indicate that ORG-AS can inhibit the transcription of *org*.

## 3. Discussion

We used RACE to clone the cDNA of the *org* gene from the ovary of *C. semilaevis*. *org* contains a 5′-cap and 3′poly-A tail, encoding 221 amino acids, including six exons and five introns. Homology and evolutionary analysis showed that *org* was only found in fish and was highly conserved. We presumed that *org* is only a MEG in fish. MEGs activate the embryonic genome by encoding maternal effector complexes, and they degrade rapidly after completing their duties, which promotes the transformation of embryos from a maternal regulatory mechanism to a zygotic regulatory mechanism [26]. In addition, maternal complexes encoded by MEGs are involved in oocyte meiosis and genomic transcriptome activation, and they promote oocyte development [3]. Then, we analyzed the expression pattern and cellular localization of *org*. The qRT-PCR results showed that *org* was highly expressed in unfertilized eggs, and after fertilization, the expression abundance decreased significantly after the blastocyst stage. Previous studies have shown that the activation of zygote genes usually occurs around the blastocyst stage in bony fish [27]. These findings are consistent with the expression pattern of *org* in early embryos. The results indicate that *org* may play a key role in the development of early embryos and MTZ.

In sexually mature *C. semilaevis*, the qRT-PCR results showed that *org* was relatively abundant in the gonads, especially in the ovary. The significant difference in the expression of *org* between the ovary and testis indicates that *org* may play an important role in the ovary. In addition, the qRT-PCR results showed that *org* was barely expressed in the early stage of ovarian development. Then, *org* was highly expressed at the beginning of reproductive cell maturation into the division period. According to the FISH results, *org* was mainly localized in the cytoplasm of oocytes. This finding is consistent with the expression patterns of the MEGs *vasa*, *zorg*, *npm2*, and *foxr1* in zebrafish [9,25,28,29]. According to the development of oocytes in the ovary, fish ovaries can be divided into three categories: the complete synchronous spawning type, the batch synchronous spawning type, and the asynchronous spawning type. The ovary of *C. semilaevis* belongs to the asynchronous spawning type; the ovary contains oocytes of different developmental stages [30]. This indicates the continuity of oogenesis in *C. semilaevis*. Liu et al. also revealed that the oogenesis of *C. semilaevis* is continuous by coordinating different granulosa cells through single-cell sequencing [31]. Previous studies have shown that the oogenesis of bony fish begins with the differentiation and proliferation of oocytes. Oocytes undergo meiosis and differentiation to form primary oocytes and secondary oocytes, which gather in the ovarian matrix, manage the stage of primary oocytes and secondary oocytes, and finally form eggs [32]. Ma et al. showed that the differentiation of *C. semilaevis* oocytes begins at 150 dpf. At this time, the primary oocytes begin to disperse in the reproductive epithelium, indicating that some reproductive cells in the ovary begin to enter mature division, marking the beginning of oogenesis [30]. These findings are consistent with the expression pattern of *org* during ovarian development. The above results suggests that *org* might function as a MEG and maintain the continuity of oogenesis in the ovary.

In addition, we found that there was a 150 bp reverse complementary sequence between lncRNA ORG-AS and *org* exon 4, and we predicted the interaction between them by IntaRNA and RNAup. To study the regulatory effect of ORG-AS on *org*, we analyzed the expression patterns and cellular localization of *org* and ORG-AS. During the whole embryonic development period, a high level of *org* was accompanied by a low expression of ORG-AS. Intriguingly, we observed the decrease of *org* and ORG-AS in the gastrula stage. Both *org* and ORG-AS were hardly detectable during the early stage of gonad development. The expressions of *org* and ORG-AS increased rapidly after the onset of oogenesis. More importantly, ORG-AS was expressed earlier than *org* in the testis and ovary. These results indicate that the expression of ORG-AS might play an important role in regulating the precise level of *org*. FISH showed that ORG-AS colocalized with *org* in oocytes. Furthermore, the overexpression of ORG-AS suppressed the expression level of *org*. LncRNA, as an important epigenetic regulator, plays an important role in gene transcription and post-transcriptional regulation [33]. Previous studies have shown that lncRNAs can not only regulate gene expression through *cis*-regulation but can also function in *trans*-regulation by interacting with proteins or RNA molecules [34]. This indicates that the *org* gene expression was regulated by the *trans*-regulation function of lncRNA ORG-AS. Our results suggest that high levels of ORG-AS maintain low expression of *org* and promote the testis development. In the ovary, ORG-AS might bind to the mRNA of *org* and regulate oogenesis.

In summary, we first identified the MEG *org* in *C. semilaevis* and cloned the full-length cDNA of *org*. The evolutionary analysis showed that *org* may be a specific MEG in fish. In the early embryonic development of *C. semilaevis*, *org* was involved in MTZ as a MEG to ensure the stability of MTZ. During oogenesis, *org* was highly expressed and mainly localized in the oocyte cytoplasm. According to the localization of *org* in oocytes, it can be used as a reproductive development marker gene in the oocytes of *C. semilaevis*. Second, we identified the lncRNA ORG-AS, determined its expression pattern and location, and found that it inhibited *org* expression through *trans*-regulation. However, the mechanisms of *org* in MTZ and oogenesis are still unclear and need further study.

## 4. Materials and Methods

### 4.1. Experimental Fish and Tissue Sampling

The Chinese tongue sole animals used in this experiment were all from Huanghai Aquiculture, Ltd. (Yantai, China). The “Guidelines for Experimental Animals” of the Yellow Sea Fisheries Research Institute (YSFRI) (Qingdao, China) were used to collect and process animals. The anatomical collection of the gonads, kidney, intestine, liver, spleen, heart, brain, skin, gill, and stomach of adult (360 days post-fertilization) *C. semilaevis* was performed. An anatomical analysis of the gonads of *C. semilaevis* at 180–720 dpf and an anatomical analysis of body parts that contained gonads at 30, 60, 90, and 120 dpf were performed. The early embryos of *C. semilaevis* were collected at the unfertilized stage, cell differentiation stage, blastocyst stage, proembryo stage, and somite stage. Each anatomical tissue was frozen in liquid nitrogen and stored at −80 °C. A caudal fin sample was collected from each fish; stored in ethanol for the purposes of obtaining genomic DNA and identifying their sex; and then classified according to stage, sex, and tissue.

### 4.2. Cloning of org

TRIzol was used to extract the RNA from each tissue of adult *C. semilaevis* and early embryos and dissolve RNA in DEPC water. RNA purification and concentration were determined using a NanoDrop2000 spectrophotometer (Thermo, Waltham, MA, USA). The integrity of the product was detected by 1% agarose gel electrophoresis, and qualified RNA was stored at −80 °C. Then, a Prime Script™ II 1st Strand cDNA Synthesis Kit (TaKaRa, Shiga, Japan) was used for cDNA transcription and synthesis. A lysis method was used to extract genomic DNA from the caudal fin and to identify fish sex using methods and primers recorded in the literature (sex-F and sex-R, Table 1) [35].

With the NCBI database, primers (*org*-F, *org*-R; Table 1) were designed according to predicted sequences (Accession No. XM_008307072.3), and PCR was performed using the gonad cDNA as the template for the amplification of *org* with Repaid DNA polymerase (TaKaRa, Shiga, Japan). In PCR, the following conditions were used: 94 °C for 5 min, followed by 35 cycles of 94 °C for 15 s, 55 °C for 15 s, and 72 °C for 5 s, and then a 72 °C extension for 5 min. A FastPure^®^ Gel DNA Extraction Mini Kit was used to purify the PCR product (Vazyme, Nanjing, China), and the target fragment was ligated into the *pEASY*-T1 cloning vector (TransGene, Beijing, China). Positive clones were screened and sequenced by Ruibo, Ltd., (Beijing, China) after being transformed into DH5α competent cells (TaKaRa, Shiga, Japan).

After confirming the sequence, we used a SMARTer RACE 5′/3′ Kit to amplify the full length of *org* (Clontech, Mountain View, CA, USA). Long-end primers in the kit and gene-specific primers (5′GSP, 3′GSP; Table 1) were used for the first round of PCR amplification. The products of the second round of amplification were amplified using the short-end primers in the kit and gene-specific primers (5′NGSP, 3′NGSP; Table 1). PCR products were transformed, linked, and sequenced as described above.

### 4.3. Real-Time Quantitative PCR

Primers were designed according to the *org* and ORG-AS sequences, and *Cs-18S-*F, *Cs-18S-*R, *β-actin*-Q-F, and *β-actin*-Q-R were used as internal reference genes (*org*-Q-F, *org*-Q-R, ORG-AS-Q-F, ORG-AS-Q-R, *Cs-18S*-F, *Cs-18S*-R, *β-actin*-Q-F, *β-actin*-Q-R; Table 1). The cDNA of each tissue was used as a template for qRT-PCR amplification to determine the expression of *org* and ORG-AS in the tissue. With a reaction volume of 20 μL, qRT-PCR was performed on a rapid real-time PCR system, the ABI 7500, real-time fluorescent quantitative PCR, Lightcycler 480II (Basel, Switzerland), using the QuantiNova™ SYBR Green PCR Kit (Qiagen, Hilden, Germany). For qRT-PCR, the following conditions were used: 95 °C for 2 min, 40 cycles of 95 °C for 5 s, and 60 °C for 10 s. The reaction conditions for the melting curve were 95 °C for 5 s, 60 °C for 1 min, +1 °C/min, and 95 °C for 15 s. Three biological and three technical repeats were required for the qRT-PCR of each tissue. Relative quantitative analysis using the 2^−ΔΔCt^ method and the normalization of *org* and ORG-AS expression using *β-actin* and *Cs-*18S rRNA were performed. For an analysis of the data, GraphPad Prism 6 was used, along with a *t* test to assess statistical significance (GraphPad Software, La Jolla, CA, USA).

### 4.4. Fluorescence In Situ Hybridization of org and ORG-AS

The ovaries at 6 mpf and 1.5 ypf were stored in 4% PFA, fixed in paraffin wax, and sliced into 3 μm sections. *org* and ORG-AS-specific fluorescent in situ hybridization probes were designed; *org* was assessed using FISH with a 5′Cy3-labeled fluorescent probe, and ORG-AS was assessed using FISH with a 5′6-FAM-labeled fluorescent probe (Table 1). Fluorescence in situ hybridization was performed with an RNA FISH kit (Gene Pharma, Shanghai, China) according to the manufacturer’s requirements. The images were acquired with a laser confocal microscope (Olympus, Tokyo, Japan) and analyzed.

### 4.5. Cell Culture, Vectors, and Transfection

ORG-AS and *org* overexpression vectors were established to confirm the role of ORG-AS. The primers shown in Table 1 were used for PCR amplification of the *ORF* CDS region and the binding site of ORG-AS. The ORG-AS and pcDNA3.1+ vectors (Invitrogen, Carlsbad, CA, USA) were cut by EcoRI and XbaI, and the *org* and pEGFP-n3 vectors were cut by EcoRI and KpnI. pcDNA3.1-ORG-AS and pEGFP-n3-*org* were ligated with T4 DNA ligase (NEB, Inc., Ipswich, MA, USA). The connected plasmids were transformed into *Escherichia coli*, pcDNA3.1-ORG-AS was coated on the plate containing ampicillin, and pEGFP-n3-*org* was coated on the plate containing Kana. After 12 h, a positive clone was selected for colony PCR, and the positive clones were sent for sequencing to verify whether the recombinant plasmids were connected correctly. The recombinant plasmid was extracted using an EndoFree Mini Plasmid Kit II (Tiangen Biotech, Beijing, China) for subsequent transfection experiments.

The 293T cell line was purchased from the Shanghai Institute of Cell Biology. L-15 Medium Leibovitz (Thermo Fisher, Carlsbad, CA, USA) was used to culture the cells at 37 °C with 5% CO2. Cells logarithmically grew to 80–90% density in medium, were digested with 0.25% trypsin, and were transferred to 24-well cell culture plates. Then, pEGFP-n3-*org* and pcDNA3.1-ORG-AS vectors were cotransfected into 293T cells using Lipofectamine™ 3000 (Thermo Fisher, Invitrogen, Carlsbad, CA, USA). The negative control group comprised pEGFP-n3-*org* and pcDNA3.1 vector-cotransfected 293T cells, and the control group comprised pEGFP-n3-*org* vector-transfected 293T cells. Cells were collected 48 h after transfection, and RNA was extracted as mentioned above. The expression of *org* was detected using qRT-PCR.

### 4.6. Sequence Analysis

RNA/RNA binding sites were predicted using IntaRNA (accessed on 13 March 2022, http://rna.informatik.uni-freiburg.de/IntaRNA/Input.jsp) and RNAup (accessed on 13 March 2022, http://rna.tbi.Univie.ac.at//cgi-bin/RNAWebSuite/RNAup.cgi). Gene homology searches were performed using the Blast tool in NCBI (accessed on 7 April 2022, http://www.ncbi.nlm.nih.gov/BLAST/). Multiple sequence alignment was performed using DNAMAN (accessed on 4 June 2022, http://www.clustal.org/omega/). A NJ phylogenetic tree was constructed using the MEGA maximum likelihood method.

### 4.7. Statistical Analysis

Experimental data are shown as the mean ± standard error and were obtained in triplicate with three independent experiments. The data were analyzed using *t* tests and plotted with the GraphPad Prism 6 software package (GraphPad Software, San Diego, CA, USA). *p* values of ≤0.05 were considered statistically significant (* *p* < 0.05, ** *p* < 0.01, *** *p* < 0.001).

## 5. Conclusions

Our study showed that *org*, as a MEG only present in fish, plays an important role in the early embryonic development and oogenesis of *C. semilaevis*. The lncRNA ORG-AS negatively regulates *org* expression through *trans*-regulation. These studies broaden our knowledge of the function of MEGs in embryonic development and oogenesis and further enrich the knowledge of lncRNAs in regulating genes.

## Figures and Tables

**Figure 1 ijms-23-08605-f001:**
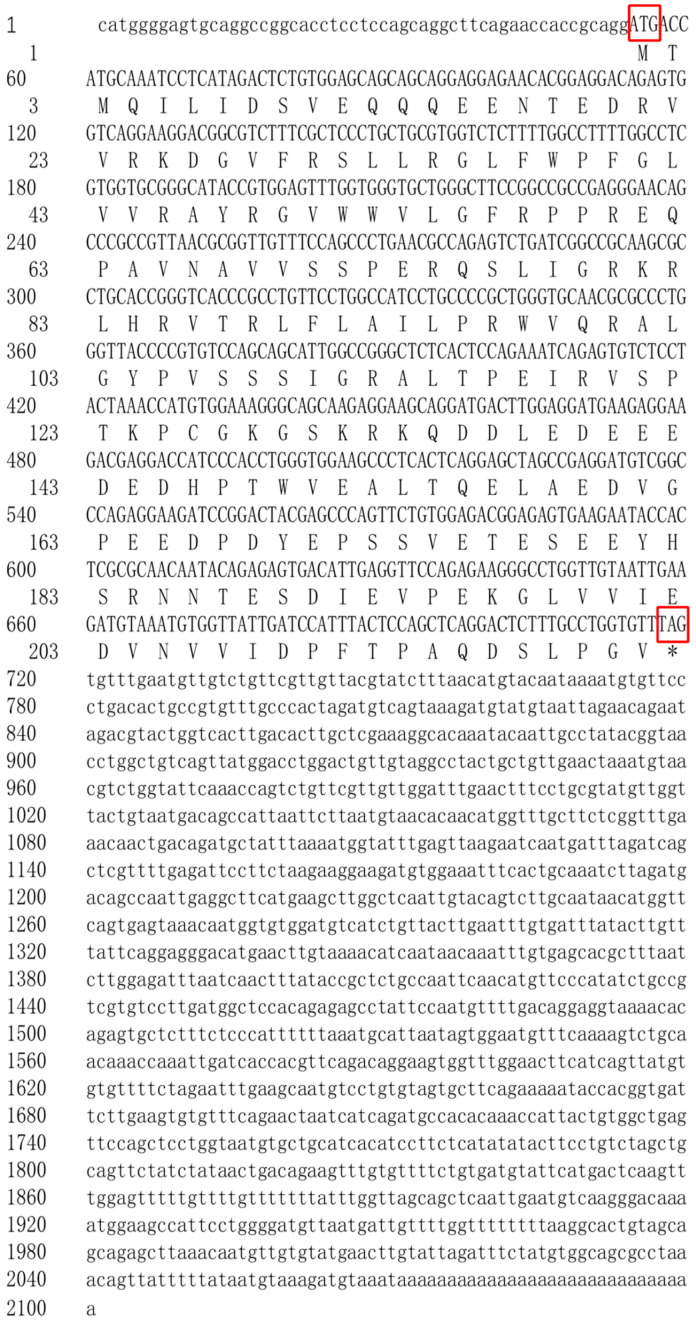
Full-length cDNA sequence and amino acid sequence of *org* gene. The ORF of *org* cDNA encoded 221 amino acids. The amino acid sequence is in uppercase letters. The red box represents the start codon and stop codon of *org*, and the stop codon is indicated by an asterisk (*).

**Figure 2 ijms-23-08605-f002:**
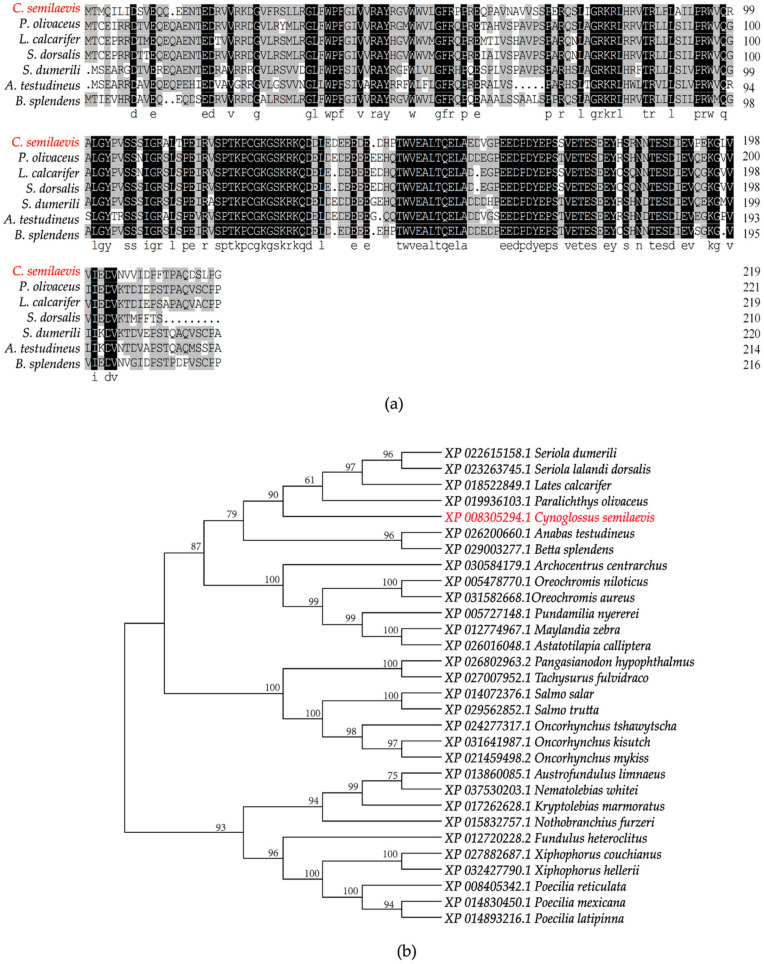
Homology analysis and phylogenetic tree of ORG proteins in bony fish. (**a**) DNAMAN was used to perform multisequence alignment of *org* amino acid sequences. Black boxes indicate sequence identity between amino acid sequences, while gray boxes indicate more than 75% similarity between amino acid sequences. Note: *S. dumerili* (Sequence ID: XP_022615158.1), *L. calcarifer* (Sequence ID: XP_018522849.1), *S. dorsalis* (Sequence ID: XP_023263745.1), *P. olivaceus* (Sequence ID: XP_019936103.1), *A. testudineus* (Sequence ID: XP_026200660.1), and *B. splendens* (Sequence ID: XP_029003277.1); (**b**) Phylogenetic tree based on the *org* amino acid sequence, using the neighbor-joining method. The number displayed on the branch node represents the guiding value (%). The *C. semilaevis org* is marked in red color.

**Figure 3 ijms-23-08605-f003:**
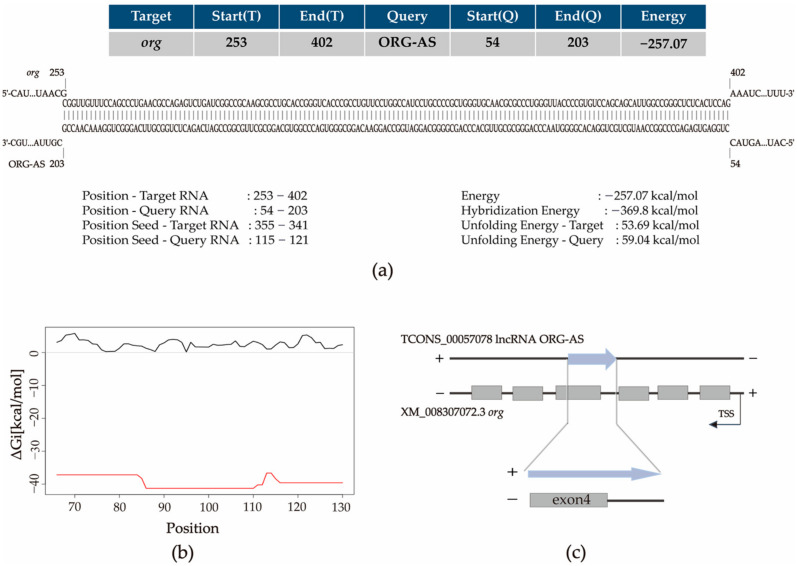
*org* and ORG-AS interaction site diagram. (**a**) IntaRNA results showed ORG-AS binding sites to *org*; (**b**) RNAup showed the interaction between nucleotides 86–110 of ORG-AS and nucleotides 348–372 of *org*. ORG-AS is represented by a black line; *org* is represented by a red line; (**c**) Diagram of ORG-AS and *org* interactions.

**Figure 4 ijms-23-08605-f004:**
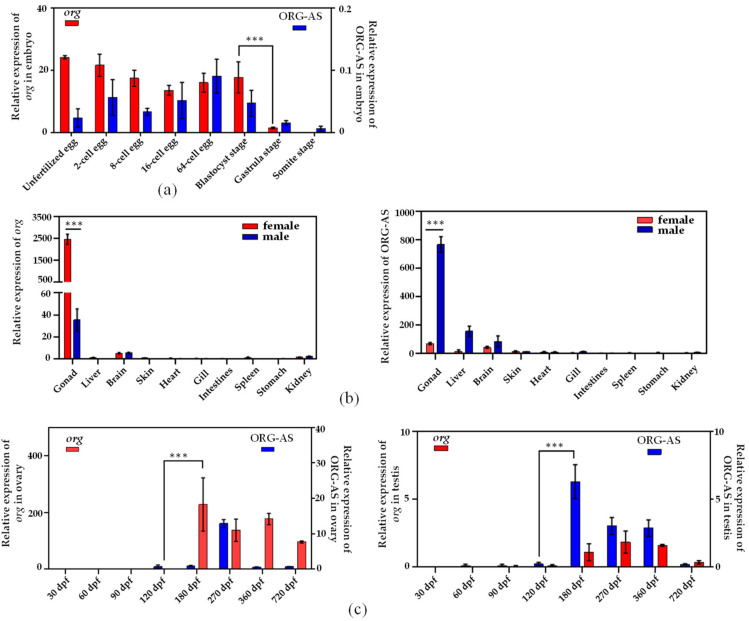
Expression of *org* and ORG-AS in *C. semilaevis*. (**a**) *org* and ORG-AS expression patterns in early embryos, and *org* and ORG-AS expression levels normalized to 18S rRNA expression; (**b**) Expression of *org* and ORG-AS in sexually mature *C. semilaevis*, and *org* and ORG-AS expression levels normalized to *β-actin* expression; (**c**) Expression patterns of *org* and ORG-AS during gonad development in *C. semilaevis*, and *org* and ORG-AS expression levels normalized to *β-actin* expression. The mean ± SEM from three independent individuals (*n* = 3) is shown. (*** *p* < 0.001).

**Figure 5 ijms-23-08605-f005:**
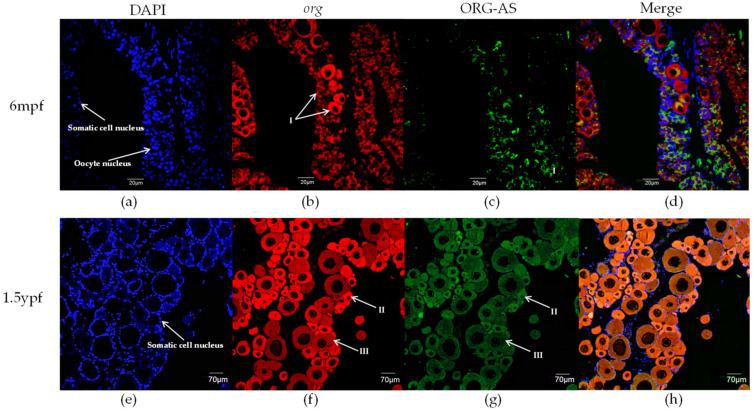
*org* and ORG-AS localization in the ovary of *C. semilaevis* at 6 mpf and 1.5 ypf. (**a**) Oocyte nuclei and somatic nuclei were stained with DAPI in 6 mpf ovaries; (**b**) Localization of *org* detected using 5′Cy3-labeled fluorescent probe in 6 mpf ovaries; (**c**) Localization of ORG-AS detected using 5′6-FAM-labeled fluorescent probe in 6 mpf ovaries; (**e**) Somatic nuclei were stained with DAPI in 1.5 ypf ovaries; (**f**) Localization of *org* detected using 5′Cy3-labeled fluorescent probe in 1.5 ypf ovaries; (**g**) Localization of ORG-AS detected using 5′6-FAM-labeled fluorescent probe in 1.5 ypf ovaries; (**d**,**h**) Combination of *org*, ORG-AS, and nucleus localization. I: phase I oogonia; II: phase II oocyte; III: early phase III oocyte.

**Figure 6 ijms-23-08605-f006:**
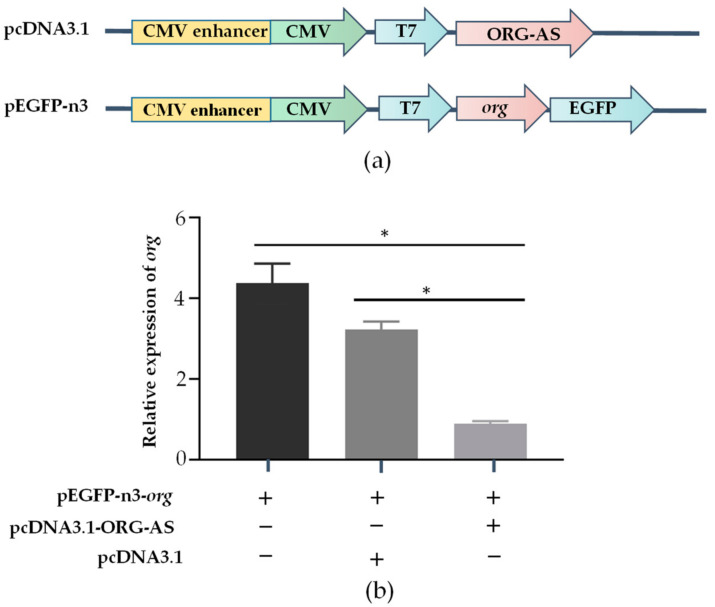
The effect of ORG-AS overexpression on *org* expression levels. (**a**) Construction of pcDNA3.1-ORG-AS and pEGFP-n3-*org* overexpression vectors; (**b**) After transfection of 293T cells, the expression level of *org* was detected and normalized to ACTINB expression, and the mean ± SEM from three independent individuals (*n* = 3) is shown. * *p* < 0.05.

**Table 1 ijms-23-08605-t001:** Primers used for cloning and gene expression analysis.

Primer Name	Sequence (5′-3′)
*org*-F	CCAGCAGGCTTCAGAACCACCG
*org*-R	CAGTAACCAACATACGCAGGAA
*org*-3-GSP	CCCTGACACTGCCGTGTTTGCCCACTAGA
*org-*3-NGSP	ACTGGTCACTTGACACTTGCTCG
*org*-5-GSP	AGCCCAGCACCCACCAAACTCCACGGT
*org*-5-NGSP	CCACTCTGTCCTCCGTGTTCTC
*org*-Q-F	CAGCAGGAGGAGAACACGGA
*org*-Q-R	ACCAAACTCCACGGTATGCC
ORG-AS-Q-F	CACTTCGGAGAACACTGGCG
ORG-AS-Q-R	TGGGGCTCTTCTATTGGTCG
*β-actin*-Q-F	GCTGTGCTGTCCCTGTA
*β-actin*-Q-R	GAGTAGCCACGCTCTGTC
*Cs-*18S rRNA-F	GGTCTGTGATGCCCTTAGATGTC
*Cs-*18S rRNA-R	AGTGGGGTTCAGCGGGGTTAC
hACTINB-Q-F	GATGATATCGCCGCGCTCGT
hACTINB-Q-R	GTAGATGGGCACAGTGTGGGTG
*org*-EcoRI-F	CGGAATTCATGACCATGCAAATCCTC
*org*-KpnI-R	CGGGTACCAACACCAGGCAAAGAGT
ORG-AS-KpnI-F	CGGAATTCCTGGAGTGAGAGCCCGG
ORG-AS-EcoRI-R	GCTCTAGAACGCTTGGTTTGAACTTGATG
*org*-5′Cy3	TGGGCTCGTAGTCCGGATCTTCCTCTGGGCC
ORG-AS-5′6-FAM	CCTCATAGACTCTTCAGGACAGAGTGGTCAGGAAGG
sex-F	CCTAAATGATGGATGTAGATTCTGTC
sex-R	GATCCAGAGAAAATAAACCCAGG

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
