# Peer review of "Cloning of the Maternal Effector Gene *org* and Its Regulation by lncRNA ORG-AS in Chinese Tongue Sole (*Cynoglossus semilaevis*)"

_ijms, 2022, doi:10.3390/ijms23158605_

Round 1

Reviewer 1 Report

 Oocyte development, maternal-zygote transition and the maintenance of the maternal transcriptome stability are key steps in the viability of animal reproduction. Studying oogenesis-related genes and those modulating all these processes are of great relevance specially in fish, where surviving rates of earliest stages of development are so low. The experimental design is concrete and concise, but the manuscript should not be accepted in this format due to serious concerns in morphology and writing (major revision). As a resume, although the results are interesting, the pathways in which genes are involved are poorly described or unconnected across the manuscript so the relevance of the results goes unnoticed. Main concerns are described below.

Title: maybe, it would be simpler to read: Cloning of the maternal effector gene org and its regulation by lncRNA ORG-AS in Chinese tongue sole (Cynoglossus semilaevis)

I strongly encourage the authors to deeply revise the manuscript regarding abbreviations’ description as I have pointed to some examples only in the first lines. As matter of fact, MEGs were initially described, then used the abbreviation and then described again, changing indistinctly across the manuscript.  Some examples:

Line 15: it should say “…(MTZ) and participate…”

Line 16: LncRNAs abbreviation should be detailed

Line 20: as org description name appeared before, it should not be described again

Line 22: what is ORG-AS?

Line 30: “at transcriptional level”

The manuscript should be reviewed by a scientific English expertise or native speaker.

In general, introduction is difficult to understand and follow due to so many genes that are not properly described (probably due to the extension that would be required) modulating many different activities that are scarcely described and connected to each other’s. It is recommended to find a simpler manner to describe the processes without introduce too much unconcise information, maybe avoiding detailing all the genes.

Line 61: the description of the oogenesis is completely inaccurate.

Discussion:

The discussion point, although more consciously written, better detailed and connected, as most of the genes were not described (function) or poorly introduced in the pathways that are involved in, or even the pathways were not fully connected in the introduction part or the discussion, it is not easy to establish a big picture of the actual relevance of the results obtained (which are very interesting).

Line 255: are there other studies in fish or other vertebrates describing the differences between ovary and testis regarding org gen expression as in this work?

Reviewer 2 Report

The article has the merits of a significant scientific and applied work, but some corrections are needed before printing.

Recommendations to authors:

1. Please better expose the purpose of the study to match the title of the article!

2. For better visualization, please adjust the scale of fig. 1 or divide it into several figures.

3. Please zoom in on figures 3 and 5 for better visualization!

4. In the "Material and method" section, please describe the equipment you used for Real-time quantitative PCR!

5. I recommend the continuation of these interesting scientific-applied biochemical-genetic studies in other species of economically significant fish!
